# Parylene-Coated Polytetrafluoroethylene-Membrane-Based Portable Urea Sensor for Real-Time Monitoring of Urea in Peritoneal Dialysate

**DOI:** 10.3390/s19204560

**Published:** 2019-10-20

**Authors:** Min Park, JeeYoung Kim, Kyounghee Kim, Jae-Chul Pyun, Gun Yong Sung

**Affiliations:** 1Major in Materials Science and Engineering, Hallym University, Chuncheon, Gangwon-do 24252, Korea; minpark@hallym.ac.kr (M.P.); jyoung_96@naver.com (J.K.); seoulhee92@naver.com (K.K.); 2Cooperative Course of Nano-Medical Device Engineering, Hallym University, Chuncheon, Gangwon-do 24252, Korea; 3Integrative Materials Research Institute, Hallym University, Chuncheon, Gangwon-do 24252, Korea; 4Department of Materials Science and Engineering, Yonsei University, Seoul 03722, Korea; jcpyun@yonsei.ac.kr

**Keywords:** urease immobilization, chemical cross-linking, surface modification, parylene-A, flow system, real-time monitoring

## Abstract

A portable urea sensor for use in fast flow conditions was fabricated using porous polytetrafluoroethylene (PTFE) membranes coated with amine-functionalized parylene, parylene-A, by vapor deposition. The urea-hydrolyzing enzyme urease was immobilized on the parylene-A-coated PTFE membranes using glutaraldehyde. The urease-immobilized membranes were assembled in a polydimethylsiloxane (PDMS) fluidic chamber, and a screen-printed carbon three-electrode system was used for electrochemical measurements. The success of urease immobilization was confirmed using scanning electron microscopy, and fourier-transform infrared spectroscopy. The optimum concentration of urease for immobilization on the parylene-A-coated PTFE membranes was determined to be 48 mg/mL, and the optimum number of membranes in the PDMS chamber was found to be eight. Using these optimized conditions, we fabricated the urea biosensor and monitored urea samples under various flow rates ranging from 0.5 to 10 mL/min in the flow condition using chronoamperometry. To test the applicability of the sensor for physiological samples, we used it for monitoring urea concentration in the waste peritoneal dialysate of a patient with chronic renal failure, at a flow rate of 0.5 mL/min. This developed urea biosensor is considered applicable for (portable) applications, such as artificial kidney systems and portable dialysis systems.

## 1. Introduction

Urea is a compound synthesized from ammonia in the liver during the decomposition of proteins, and it represents the final nitrogenous end product of metabolism [1]. Urea is widely used in conjunction with creatinine as an important marker of renal function [2]. The normal ranges of urea and creatinine in blood are 7–20 mg/dL and 0.7–1.2 mg/dL, respectively [3,4]. In renal failure, the glomerular filtration rate drastically drops, which results in increased concentrations of urea and creatinine in serum. If chronic kidney failure continues and becomes severe, it will progress to end-stage renal failure (ESRF) [5,6]. In ESRF, a kidney transplant must be performed, and hemodialysis or peritoneal dialysis is required until the transplant is carried out [7]. The concentrations of urea and creatinine in the blood or peritoneum are the major markers of the progress of dialysis [8].

Various assays and biosensors based on electrochemical, thermal, optical, and piezoelectric detection have been developed for monitoring the concentration of urea [9,10,11]. Among these, electrochemical urea biosensors have been widely developed because of their high sensitivity and their efficient and rapid assay [12]. For the electrochemical measurement of the concentration of urea, a urease-based enzyme biosensor has been developed [13]. Urease is a nickel-containing metalloenzyme found in various bacteria, fungi, algae, and plants [14]. Urease specifically reacts with urea, so a urease-based biosensor can minimize the interference from other molecules in physiological fluid, such as uric acid and glucose [15,16]. 

In previous studies, urease was immobilized on the surface of an electrode that was used for electrochemical measurements [11,17]. However, direct immobilization limits the area of urease immobilization to the area of the electrode and increases the cost of the assay, because urease must be re-immobilized each time the electrode is replaced [18]. As a better alternative, nanostructures onto which urease could be immobilized were constructed on the surface of an electrode [11,19]. However, this type of biosensor also requires the re-immobilization of urease after each replacement of the electrode [20]. Recently, urease was immobilized on a separate substrate that was tightly attached to an electrode, and the concentration of urea was measured electrochemically; urease was immobilized on a porous substrate to increase the area and improve the sensitivity [21,22]. However, in these studies, urea measurements were performed in the static condition. The measurement of urea concentration in the static condition after the collection of blood or urine is useful for medical diagnosis, but in order to monitor the progress of dialysis during hemodialysis or peritoneal dialysis urea needs to be measured in the flow condition. Thus, urea measurement in the flow condition has a clear and irreplaceable advantage, real-time monitoring. The use of a bioreactor for flow analysis was reported recently [23]. However, this bioreactor required a long reaction time and measured only a single light signal from a single injection of urea after a complex transferring process. Therefore, it is not considered suitable for the continuous monitoring of urea in physiological samples. Ohnishi et al. measured urea concentration using a microfluidic chip [24]. Although this device utilized the flow channel, it measured the potential in the static condition and the device could be used only for a single measurement. A thermal biosensor has also been utilized for flow-injection analysis [25]. However, the detected urea concentration (100 mM) was much higher than the normal range and a continuous monitoring was impossible. In addition, most of electrochemical urea biosensors utilize the additional enzyme, such as glutamate dehydrogenase, for electron generation or nanostructures [20,26]. These additional precious materials or processes result in an increase in the total cost of the measurement. Very recently, we reported the continuous detection of urea in the flow condition [27]. Urease was immobilized on porous silk fibroin and urea detection was tested with various flow rates, and it was confirmed to be feasible at flow rate slower than 0.5 mL/min. Anyhow, for real-time monitoring of urea in physiological samples, a biosensor that continuously monitors urea in the fast flow condition needs to be developed.

In this study, a high density of urease was immobilized on porous membranes, and a real-time urea-monitoring biosensor was fabricated with the membranes for use in the flow condition. For simple and low-cost monitoring of urea, a screen-printed carbon electrode was used after the amination. To achieve high-density urease immobilization, the porous membranes were coated with an amine-functionalized parylene, parylene-A. Parylene (poly(*p-xylylene*)) is a polymer that can be coated on porous membranes by vapor deposition at room temperature. Parylene-A contains one amine group per repeating unit. Therefore, the porous membranes were first coated with parylene-A to achieve a high concentration of amine groups on the surface, and subsequently, a high density of urease was immobilized on the membranes using glutaraldehyde as the crosslinker. The porous membranes maximized the contact area with the fluid in the flow condition, thus improving the sensitivity of the electrochemical measurement. Finally, the urease-immobilized porous membranes were inserted into a polydimethylsiloxane (PDMS) chamber to form a fluidic system [18,28,29]. The concentration of urea was then monitored using the fabricated urea biosensor in the flow condition.

## 2. Materials and Methods

### 2.1. Materials

Urease (from Canavalia ensiformis), urea, glutaraldehyde, disodium hydrogen phosphate, potassium dihydrogen phosphate, fluorescein, fluorescein isothiocyanate (FITC), and ammonium carbamate were obtained from Merck (Darmstadt, Germany). The urea assay kit was purchased from BioAssay Systems (Hayward, CA, USA). Phosphate buffered saline (PBS) was obtained from LPS solution (Daejeon, Korea) and used as the supporting electrolyte for the electrochemical measurements. Phosphate buffer (PB) was prepared by mixing 20 mM disodium hydrogen phosphate and 20 mM potassium dihydrogen phosphate (pH = 7.4), and it was used as the immobilization buffer. Porous polytetrafluoroethylene (PTFE) membranes with a pore size of 1 μm were purchased from Advantec MFS (Dublin, CA, USA). The waste peritoneal dialysate of a patient with chronic renal failure was obtained from the Seoul National University Hospital (Seoul, Korea) with written consent, and they agreed to participate according to the Declaration of Helsinki. The study was approved by the institutional review boards of Seoul National University Hospital (protocol number: SNUH IRB No. 1610-016-797).

### 2.2. Parylene Coating of PTFE Membrane and Urease Immobilization

Typically, a porous PTFE membrane was punched using a biopsy punch (diameter = 8 mm) and a parylene film was deposited on it by the following procedure: (1) The parylene dimer functionalized with amine was evaporated at 160 °C; (2) the dimer was pyrolyzed at 650 °C to yield the highly reactive amine-functionalized p-xylene radical; (3) the amine-functionalized parylene film was uniformly deposited on the PTFE membrane at room temperature. Vacuum (<5 Pa) was maintained during the whole coating procedure. After deposition of the parylene film, the amine groups on the film surface were converted into active aldehyde groups by treatment with a 10% solution of the crosslinking agent glutaraldehyde in PB, with vigorous shaking for 1 h. Finally, urease was immobilized on the PTFE membrane via the chemical reaction between the active aldehyde groups and the free amine groups in urease. 

### 2.3. Urease Activity Assay

A commercial urea assay kit was used to measure the urea concentration. In brief, the urease-immobilized PTFE membrane was treated with 16.67 mM urea for 1 h at 25 °C, under shaking. After the urea was hydrolyzed by the urease in the urease-immobilized PTFE membrane, 5 μL of the hydrolyzed urea solution was transferred to a 96-well microplate, and treated with 200 μL of the phthalaldehyde reagent from the kit for 20 min. The activity of the immobilized urease was then measured by colorimetry by optical density measurements at a wavelength of 520 nm. All experiments were repeated three times. 

### 2.4. Fabrication of the Urea Biosensor

The urea biosensor fabricated in this study is depicted in Figure 1. The sensor comprised a lower housing, a three-electrode system, a polydimethylsiloxane (PDMS) fluidic chamber, urease-immobilized PTFE membranes (various numbers: 4, 6, 8, 10, or 12), and an upper housing. The housings were 3D-printed using acrylonitrile poly-butadiene styrene (ABS) filaments. To fabricate the PDMS fluidic chamber, PDMS was solidified on a Si wafer adhered to a plastic frame (width = 10.20 mm, length = 14 mm, and height = 5.43 mm). After solidification, a cylindrical fluidic chamber with inner diameter 8 mm was punched with the biopsy punch. The screen-printed three-electrode system comprised a 4 mm-diameter working electrode (DropSens, Llanera, Spain), which was used after amination [23]. For amination, the electrode was dipped in a 0.5 M ammonium carbamate solution, and cyclic voltammetry (CV) was performed. The sweep potential and rate were set at values ranging from 0.5 to 1.2 V and 20 mV/s for 50 cycles, respectively. After the amination, the electrode was washed with de-ionized water (DW). After all the compartments of the urea biosensor were prepared, the urease-immobilized parylene-coated PTFE membranes were inserted into the PDMS fluidic chamber, and the other parts were assembled as shown in Figure 1a.

### 2.5. Flow System Configuration and Electrochemical Measurements

A single flow system was constructed to detect the urea concentration in the flow condition, as shown in Figure 1b. The flow cell of the urea sensor was connected to the tubes, and an open circuit was created through the flow. The flow rate was controlled using a peristaltic pump (Ismatec, Wertheim, Germany) from an open circuit that was connected to the inlet tube; the outlet was connected to a waste bottle. Potentiostats (DropSens, Llanera, Spain) were connected to the electrode installed in the sensor to determine the current generated from the hydrolysis of urea by urease. For real-time monitoring of urea in the flow condition, chronoamperometry was performed at 1.1 V under various flow rates.

## 3. Results and Discussion

### 3.1. Fabrication of the Urease Immobilized PTFE Membranes Using a Parylene Coating

Electrochemical reactions occur on the surface of the electrode used; thus, in enzyme-based electrochemical biosensors the enzyme is generally immobilized on the surface of the electrode. In this type of biosensor, the enzyme and the electrode must be simultaneously replaced when the enzyme activity decreases due to repeated measurements, decreasing the lifetime of the electrode and increasing the cost of the biosensor. In addition, the enzyme immobilization area in such biosensors is limited to the area of the electrode. Additionally, the immobilization environment is determined by the electrode material, making the immobilization of the enzyme highly specific. In this study, urease was first immobilized on porous membranes with high surface areas, and subsequently, a sensor based on the urease-immobilized membranes was fabricated for monitoring urea in the flow condition. For the immobilization of urease, PTFE membranes, which have excellent chemical resistance, were selected. To generate urease-immobilization sites, parylene-A was coated on the membranes by vapor deposition. Parylene has excellent chemical resistance and mechanical properties; the chloride-group containing parylene-C has been approved by the Food and Drug Administration (FDA), as well. In this study, parylene-A was uniformly coated onto the surface of the porous PTFE membranes by vapor deposition at room temperature. Parylene-A has one amine group per repeating unit; thus, via deposition of parylene-A onto the PTFE membranes, their surfaces could be modified with amine functional groups. Subsequently, the membranes were treated with the dialdehyde crosslinker glutaraldehyde to functionalize the surfaces with aldehyde groups via the reaction of the amine and the aldehyde. Urease was then immobilized on the membranes via the reaction of the amine group of urease and the aldehyde groups on the membrane surfaces. The irreversible chemical crosslinking of urease onto the membranes is assumed to be more stable than reversible physical adsorption in the flow condition [30]. In addition, this prevention of the biodegradation means an improvement in the durability of urease immobilization. For the long-term usage, the stability of the urease immobilization was essential and it was achieved by the covalent immobilization in this study. Thus, the adopted strategy prevented the biodegradation of urease and improved the durability of the urea sensor for long-term usage. Amine functionalization by the parylene-A coating was analyzed using fluorescence microscopy, as shown in Figure 2. Here, 100 mg of the parylene dimer was used to obtain a 100 nm-thick coating on the porous PTFE membranes. The amine groups were visualized using FITC via the specific reaction between the amine and isothiocyanate groups. As shown in Figure 2a, no fluorescence was observed when an unfunctionalized parylene-N-coated PTFE membrane was treated with 1 μg/mL of FITC. However, a weak fluorescence (Figure 2b) was observed from the FITC-treated bare porous PTFE membrane, which could be attributed to the non-specific binding between FITC and PTFE. The PTFE membrane, because of its hydrophilic nature, binds in a non-specific manner. However, parylene, which is a hydrophobic polymer, blocks the non-specific binding; therefore, strong and uniform fluorescence was observed when the parylene-A-coated PTFE (AP) membrane was treated with FITC (see Figure 2c). In contrast, when unfunctionalized fluorescein was applied to the AP membrane, no fluorescence signal was detected (see Figure 2d). Thus, via a comparison of Figure 2b with Figure 2a,d, it was confirmed that the parylene coating prevented non-specific binding. This indicates that the parylene coating could prevent the binding of various biomolecules, injected in the flow, to the urease-immobilized AP (UAP) membranes, reducing the noise of the biosensor. In addition, a comparison of Figure 2c with Figure 2a,d demonstrated that only the amine-containing parylene layer and ITC-conjugated fluorescein reacted with each other, and no reaction took place between parylene-N and FITC or between parylene-A and fluorescein. This indicates that ITC is an amine-specific binding group, and that the parylene-A coating uniformly functionalized the surface of the PTFE membrane with a high density of active amine groups, and blocked non-specific binding.

The urease-immobilized PTFE membrane was observed using a scanning electron microscope (SEM) with 5000-fold magnification. The microporous structure of the PTFE membrane (Figure 3a) was retained after the vapor deposition of parylene-A (Figure 3b). In addition, after treatment with glutaraldehyde (Figure 3c) and urease immobilization (Figure 3d), the microstructure of the PTFE membrane was unchanged. Thus, the vapor deposition of parylene-A on the PTFE membrane did not affect its microporous structure. In addition, SEM characterization confirmed the formation of a uniform parylene-A layer on the PTFE membranes. Moreover, the chemical treatments with the crosslinker and urease did not affect the microporous structure of the PTFE membranes. The atomic distribution was analyzed using energy dispersive spectrometry (EDS) in SEM. As shown in Table 1, bare PTFE membrane showed only carbon (40.7%) and fluorine (59.3%). PTFE has four fluorine on a saturated alkane chain so it is valid result. After parylene-A coating (AP membrane), nitrogen (9.8%) appeared, and this nitrogen was from amine groups on parylene. When the AP membrane was treated with a cross-linker, glutaraldehyde, additional oxygen (1.9%) was observed. This means that AP membranes reacted with oxygen containing glutaraldehyde. After urease immobilization, nitrogen and oxygen contents were increased to be 9.7% and 4.4%, respectively. Since proteins like urease contain relatively higher amount of nitrogen and oxygen than organic polymers, this result means that urease was immobilized on the AP membrane via glutaraldehyde cross-linking. Based on these results, the fabrication of UAP membranes with an intact microporous structure and urease immobilization on AP membranes was confirmed. Because the UAP membranes were to be packed in the PDMS flow chamber, the retention of the porous structure was essential for the flow of urea samples.

The change in the functional groups on the membrane surface during the urease immobilization was analyzed using Fourier-transform infrared (FTIR) spectroscopy. As shown in Figure 4a, the bare-PTFE-membrane spectrum exhibited two CF_2_ stretching peaks, at 1205 cm^−1^ and 1150 cm^−1^. In the spectrum recorded after the membrane was coated with parylene-A, additional peaks assigned to two NH_2_ stretches, N–H bending, and aromatic C=C stretching were observed at 3443, 3375, 1622, and 1513 cm^−1^, respectively (Figure 4b). Thus, the coating of the PTFE membrane with parylene-A, with amine groups on its phenyl group-containing backbone (*p-xylylene*), was confirmed [31]. When the AP membranes were treated with glutaraldehyde, the amine peak disappeared and new C–H stretching peaks corresponding to the aliphatic groups appeared at 2919 cm^−1^ and to the aldehyde groups appeared at 2850 cm^−1^ (Figure 4c), indicating that the amine group of the parylene-A coating reacted with the aldehyde group of glutaraldehyde, and the surface of the parylene-A-coated-PTFE membrane was functionalized with aldehyde groups. The thickness of the parylene-A layer was 100 nm, which was much lower than the IR penetration depth of several micrometers [32]. Therefore, the CF_2_ stretching peaks were much more intense than the amine or aldehyde peaks. After urease immobilization, additional peaks due to amide A, amide B, amide I, and amide II were observed at 3270, 2925, 1652, and 1559 cm^−1^, respectively (Figure 4d). For urease immobilization, aldehyde modified surfaces form an imide bond with the amine in the urease. However, in this case, the intensities of amide peaks were strong, so an imide peak could not be detectable. These amide peaks are typical to proteins, confirming the immobilization of urease on the AP membranes. From these data, urease was confirmed to be immobilized via the following steps: (1) the surface of the porous PTFE membrane was modified with active amine groups by vapor deposition of parylene-A; (2) the surface of the AP membrane was then modified with aldehyde groups by treatment with glutaraldehyde; (3) free-amine-containing urease was immobilized on the AP membrane by chemical crosslinking. In addition, minimized non-specific binding was realized by the parylene coating (Figure 2). A urea biosensor with enhanced sensitivity and reduced noise could be realized via the use of the as-fabricated UAP membranes. The enhancement in sensitivity is attributed to maximization of enzyme immobilization area, due to the porous structure of the membranes, and the noise reduction is attributed to the non-specific binding [33].

### 3.2. Optimization of the Urea Sensor Based on Urease-Immobilized PTFE Membranes 

A urea biosensor based on the UAP membranes was fabricated as depicted in Figure 1a. To test the efficiency of urea immobilization and optimize the urease concentration for the immobilization, urease concentrations of 0, 4, 32, 48, 64, and 128 mg/mL were used. After immobilization, the activity of the immobilized urease was tested using the urease activity assay kit. A standard urea sample (16.67 mM) was added to the UAP membranes, and the concentration of the residual urea was measured after 1 h at 25 °C; the activity of the immobilized urease was calculated according to the amount of urea hydrolyzed: 1 U of urease liberated 1.0 μmol of NH_3_ per minute at pH 7.0 and 25 °C. The activity of the immobilized urease (■) increased as the concentration of urease used for the immobilization was increased (Figure 5), with a maximum activity of 137.5 mU when 48 mg/mL of urease was used. However, at still higher concentrations, the activity of the immobilized urease decreased. This indicates that treatment with 48 mg/mL urease resulted in the immobilization of the maximum amount of urease on the AP membranes through glutaraldehyde crosslinking. The use of higher urease concentrations resulted in reduced activity of the UAP membrane due to the decrease in the urea hydrolysis epitope or the hook effect. The optimized urease concentration was tested by electrochemical measurements. UAP membranes treated with various concentrations of urease were inserted into the PDMS chamber, and 10 mM of static urea was monitored using amperometry at an applied voltage of 1.1 V [22,27]. Urease catalyzed the hydrolysis of urea into carbamic acid and ammonia. The hydrolysis of urea is given as (Figure 1c):(NH_2_)_2_CO (urea) + H_2_O → NH_2_COOH + NH_3._

Urease hydrolyzed urea into a molecule of ammonia and a carbamic acid; the unstable carbamic acid then formed a carbamic acid radical and two radicals dimerized into a carbamic acid dimer. The carbamic acid dimer then formed a hydrazine by decarboxylation. Finally, the electrode oxidation occurred by the generation of free nitrogen from the hydrazine [34]. The intensity of the signal was proportional to the amount of hydrolyzed carbamic acid molecules present, making urease-based biosensors capable of electrochemical quantification. To achieve high sensitivity, a high concentration of urease is required for the generation of detectable amount of carbamic acid molecules; thus, high-density immobilization of urease, especially maximized density of urease’s active site, is essential. In addition, most urea sensors utilized a secondary enzyme such as glutamate dehydrogenase and its cofactor like reduced nicotinamide adenine dinucleotide (NADH) or reduced nicotinamide adenine dinucleotide phosphate (NADPH) as an electron carrier [13,35,36]. This method required extra immobilization process of the glutamate dehydrogenase and resulted in the reduction of the urease immobilization site. Furthermore, the addition of cofactor precluded the direct monitoring of urea from the physiological fluid in flow condition. As shown in Figure 5, the current (●) increased as the concentration of urease used in the treatment increased, and the maximum value of 4.46 μA was measured for the UAP membrane obtained by treatment with 48 mg/mL urease. At still higher concentrations, the current decreased. This phenomenon can be explained by the enzyme epitope. The activity of the immobilized enzyme is not from the concentration of the density of enzyme but from the density of the enzyme epitope. For the enzyme reaction, the substrate binding epitope is relatively small and localized. This means that epitope can be covered or shielded by neighbor enzyme when the density of immobilized enzyme is too high. These electrochemical data perfectly matched the results of the urease activity assay. This indicates that the sensitivity of the urea biosensor increased as the activity of the urease immobilized on the AP membranes increased. The optimal concentration for the urease immobilization was confirmed to be 48 mg/mL by both the urease activity assay and the electrochemical measurement, and these optimized UAP membranes were used to fabricate a urea biosensor for the real-time monitoring of urea in the flow condition.

### 3.3. Real-Time Monitoring of Urea Concentration in Flow Conditions

The optimized UAP membranes were assembled in the PDMS fluidic chamber as depicted in Figure 1a. The fabricated urea sensor was tested using various numbers of UAP membranes in the fluidic chamber at a flow rate of 10 mL/min. The height of the chamber was 5.43 mm, and the maximum number of membranes that could be inserted was 12. As shown in Figure 6, 4, 6, 8, 10, and 12 UAP, membranes were inserted into the urea biosensor, and the urea samples were monitored using chronoamperometry at a constant potential of 1.1 V. In all cases except for the four-membrane configuration, the current increased an increase in urea concentration. This indicates that urea monitoring in the flow condition can be realized by chronoamperometry using six or more UAP membranes. When only four membranes were used (▲), the current decreased at urea concentrations above 8 mM. The height of the PDMS chamber was much greater than the total thickness of the four membranes; therefore, the membranes might not have been in contact with the electrode (system) due to the flow of the urea sample. Additionally, a lower amount of total-immobilized-urease was present in the four UAP membrane configuration than in the other configurations. Therefore, the decrease in the current seemed to stem from a shortage of immobilized urease and the long distance between the electrode and the membranes. The eight UAP membrane configuration (■) showed the highest current values at a given urea concentration. The current increased with an increase in the number of membranes, and the maximum value was measured for the eight membrane configuration. However, the current decreased wit further increase in the number of membranes. The current measured for the 12 membrane sensor (▼) was even lower than that measured for the six membrane sensor (●). The decrease in the current at higher numbers of UAP membranes was believed to be due to a decrease in the urea-sample flow near the electrode. When a large number of membranes were inserted, the membranes overfilled the PDMS chamber; consequently, the urea samples flowed through the fluidic chamber without permeating the porous UAP membranes. This prevented urea molecules from coming into contact with the immobilized urease, which led to a decrease in the measured current. The optimized number of UAP membranes was confirmed to be eight. Therefore, the urea sensor fabricated with eight UAP membranes was used for real-time urea monitoring.

To test the real-time monitoring of urea concentration in the flow condition, urea samples were spiked into PBS and allowed to flow into the PDMS chamber at flow rates of 0.5, 1, and 10 mL/min for 20 min; subsequently, chronoamperometry was performed. The normal range of urea in blood is 7–20 mg/dL (1.2–3.3 mM); therefore, for the measurements, the urea concentration was fixed in the range 0.6–20 mM. The urea sensor was fabricated for monitoring urea concentration in physiological samples such as blood and peritoneal fluid; therefore, the maximum urea concentration was set as 20 mM. The minimum urea concentration was set as 0.6 mM because it was approximately half of the normal minimum concentration. Thus, fixing such a small value as the lower limit is beneficial for urea monitoring during dialysis. As shown in Figure 7a, the highest current was measured at a flow rate of 0.5 mL/min (red) at all ranges, and the response time was calculated to be less than 10 min. For flow rates 1 mL/min (black) and 10 mL/min (blue), the currents measured for low urea concentrations (0.6 and 1.2 mM) were not proportional. Anyhow, the urea sensor based on UAP membranes showed clear concentration-dependent current values at the other flow rates. For the analysis of a real sample, the peritoneal fluid obtained from a patient with chronic renal failure after peritoneal dialysis was used. The urea concentration in the waste peritoneal dialysate was calculated to be 20 mM; therefore, it was diluted with PBS to make samples for monitoring. These samples were injected into the PDMS chamber at a flow rate of 0.5 mL/min and were monitored using chronoamperometry. In the real sample analysis, the minimum urea concentration was selected as 1.2 mM. As shown in Figure 7b, the urea sensor based on UAP membranes clearly showed concentration-dependent current values for the real samples. When the urea concentration was changed, the current value immediately changed and stabilized within 3 min, indicating that the response time of the UAP-based urea sensor was 3 min even in flow conditions. Such a fast response will enable efficient real-time monitoring of urea concentration in physiological fluids. The current values were plotted against the urea concentration (0.6–10 mM), as presented in Figure 7c. At all flow rates, the results showed good linearity with R-square values (>0.99). At a flow rate of 0.5 mL/min, the signals from the lowest urea concentration (0.6 mM in PBS (■) and 1.2 mM (▼) in waste peritoneal dialysate) were detectable. This means that the limit of detection (LOD) was less than 0.6 mM and 1.2 mM in PBS and waste peritoneal dialysate, respectively. In addition, the LOD corresponding to the flow rates 1 mL/min (●) and 10 mL/min (▲) was the same, i.e., 4 mM. The sensitivities corresponding to flow rates 0.5, 1, and 10 mL/min in PBS were calculated to be 4.05, 1.31, and 0.46 mA·M^−1^·cm^−2^, respectively, and for the real-sample analysis, the sensitivity (▼) was calculated to be 2.4 mA·M^−1^·cm^−2^ from the slope of the Figure 7c. Although these sensitivities were not high in comparison with those previously reported for static conditions, the fluidic measurement aims to monitor urea concentration, whether it is in the normal range (0.7–1.2 mM) or not. The average urea concentration of the waste peritoneal dialysate from the patients of the renal failure was approximately 20 mM. Thus, these values are considered suitable for monitoring the concentration of urea in the range 0.6–20 mM for clinical applications [37]. In addition, the determination of urea concentration in fast flow conditions has not been reported yet; therefore, we believe that the sensitivities and LODs obtained in this study are adequate and reasonable for the real-time monitoring. From these data, we confirmed the applicability of the fabricated urea biosensor based on UAP membranes for operation in flow conditions with suitable sensitivity. The biosensor was confirmed to be usable for urea monitoring in the concentration range 0.6–20 mM at a low flow rate of 0.5 mL/min. At flow rates higher than 1 mL/min, the dynamic range of urea biosensor was confirmed to be 4–20 mM. In addition, the fabricated urea sensor was confirmed to function in the physiological fluid, waste peritoneal dialysate, with suitable sensitivity and dynamic range in the flow condition. This developed urea biosensor is considered applicable for monitoring urea concentration during surgical operation, dialysis, or maintaining artificial kidney.

## 4. Conclusions

In this study, a portable urea sensor was fabricated based on parylene-A-coated porous PTFE membranes for monitoring urea concentration in the fast flow condition. The urea-hydrolyzing enzyme, urease, was immobilized on the amine functionalized AP membranes via glutaraldehyde crosslinking to generate a specific electrochemical signal. For biosensor fabrication, the UAP membranes were inserted into a PDMS fluidic chamber. A screen-printed carbon electrode (system) was used to apply the potential and detect the electrochemical signal. For the maximum signal generation, treatment with 48 mg/mL urease was found to be optimal and the eight membrane configuration provided the maximum current. Under the optimized conditions, urea samples were monitored at various flow rates. The sensitivities at flow rates of 0.5, 1, and 10 mL/min were calculated to be 4.05, 1.31, and 0.46 mA·M^−1^·cm^−2^, respectively. The fabricated urea biosensor was confirmed to be suitable for the real-time monitoring of urea at flow rates ranging from 0.5 to 10 mL/min. From these results, it was confirmed that urea monitoring is capable in fast flow conditions like 10 mL/min. In addition, the physiological fluid from a patient with renal failure was tested at a flow rate of 0.5 mL/min and the sensitivity was calculated to be 2.4 mA·M^−1^·cm^−2^. The developed urea sensor was less than 4 cm in width, 3 cm in depth, and 2 cm in height, and therefore, the sensor is considered suitable for (portable) applications, such as artificial kidney systems and portable dialysis systems. 

## Figures and Tables

**Figure 1 sensors-19-04560-f001:**
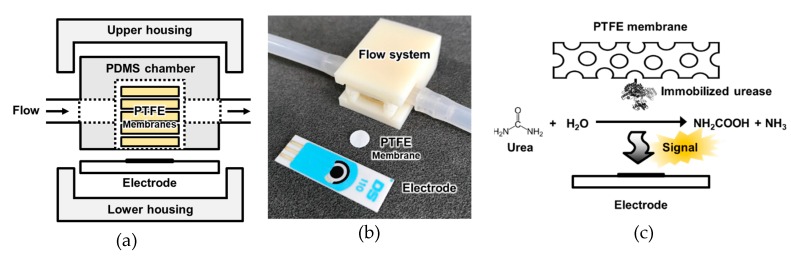
Schematic diagram of (**a**) the configuration of the sensor system; (**b**) photograph of the sensor units; and (**c**) the principle of signal generation by the urease-immobilized membranes. The urease immobilized membranes were inserted into the polydimethylsiloxane (PDMS) fluidic chamber and electrochemical signal was measured by electrode.

**Figure 2 sensors-19-04560-f002:**
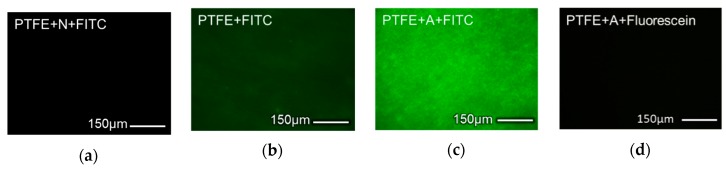
Fluorescence microscopy images of (**a**) fluorescein isothiocyanate (FITC)-treated parylene-N-coated polytetrafluoroethylene (PTFE) membrane; (**b**) FITC-treated bare PTFE membrane; (**c**) FITC-treated parylene-A-coated PTFE membrane; and (**d**) fluorescein-treated parylene-A-coated PTFE membrane.

**Figure 3 sensors-19-04560-f003:**
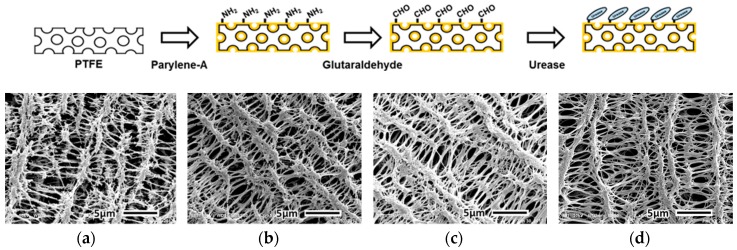
SEM (scanning electron microscope) images showing microstructures of (**a**) bare PTFE membrane; (**b**) parylene-A-coated PTFE membrane; (**c**) glutaraldehyde-treated parylene-A-coated PTFE membrane; and (**d**) urease-immobilized parylene-A-coated PTFE membrane.

**Figure 4 sensors-19-04560-f004:**
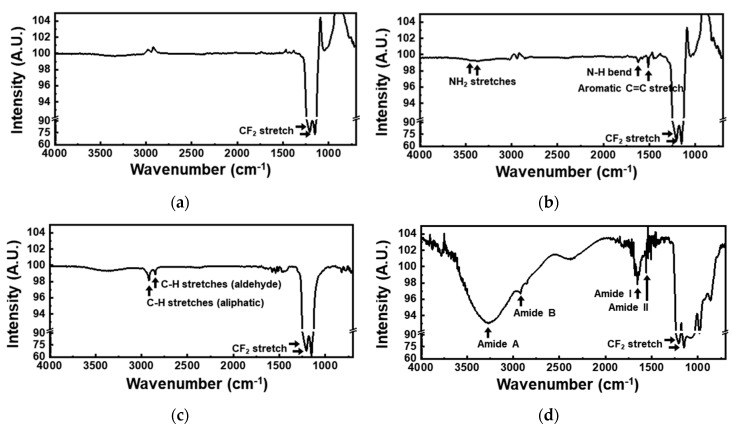
Fourier-transform infrared (FTIR) spectra of (**a**) bare PTFE membrane; (**b**) parylene-A-coated-PTFE membrane; (**c**) glutaraldehyde treated parylene-A-coated-PTFE membrane; and (**d**) urease immobilized parylene-A-coated-PTFE membrane. These spectra were measured by the sequential treatment of parylene-A, glutaraldehyde, and urease to the PTFE membrane. The spectra consisted of two parts, ranges from 4000 nm to 1300 nm and from 1300 nm to 600 nm, with different intensity ratios.

**Figure 5 sensors-19-04560-f005:**
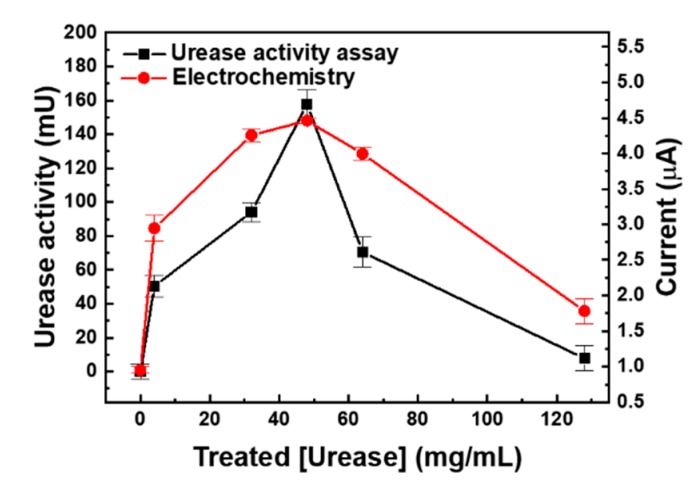
Optimization of urease concentration by urease activity assay (■) and amperometry (●). Various concentrations of urease were immobilized on AP membrane via glutaraldehyde cross-linking. The activity of immobilized urease was measured using the commercial urea assay kit and electrochemical signal was measure using fabricated urea sensor.

**Figure 6 sensors-19-04560-f006:**
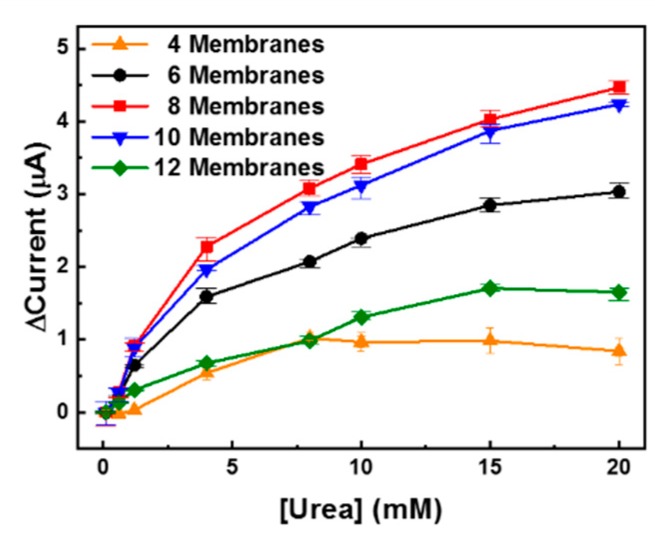
Detection of urea with various numbers of urease-immobilized parylene-A-coated PTFE (UAP) membranes. Various numbers of UAP membranes were inserted into the PDMS fluidic chamber, and the urea samples were measured to find the optimal UAP membrane numbers.

**Figure 7 sensors-19-04560-f007:**
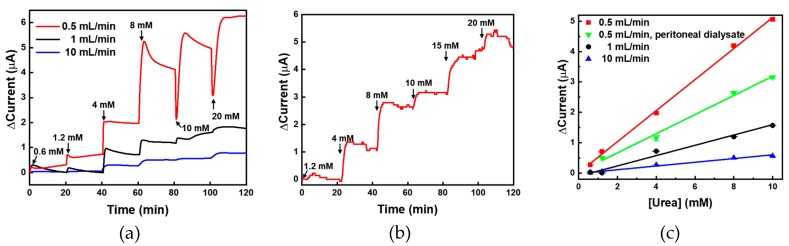
(**a**) Real-time monitoring of urea using UAP membrane-based biosensor in PBS at various flow rates. The urea samples were flow inter the UAP membrane inserted PDMS fluidic chamber and the real-time electrochemical signal was measured by the chronoamperometry; (**b**) For the analysis of real samples, the urea in human peritoneal dialysate was measured at a flow rate of 0.5 mL/min; (**c**) The urea-sensor responses were plotted against the urea concentration with the linear fitting.

**Table 1 sensors-19-04560-t001:** Atomic distribution of bare PTFE membrane, parylene-A-coated PTFE membrane, glutaraldehyde-treated parylene-A-coated PTFE membrane, and urease-immobilized parylene-A-coated PTFE membrane. Numbers in table mean atomic%.

Sample	Carbon	Nitrogen	Oxygen	Fluorine
PTFE	40.7	-	-	59.3
AP	37.7	9.8	-	52.5
AP + glutaraldehyde	38.7	8.6	1.9	50.8
UAP	35.1	9.7	4.4	50.8

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
