# Peer review of "Parylene-Coated Polytetrafluoroethylene-Membrane-Based Portable Urea Sensor for Real-Time Monitoring of Urea in Peritoneal Dialysate"

_sensors, 2019, doi:10.3390/s19204560_

Round 1

Reviewer 1 Report

The overall article is sound enough to be published, the only weak area is dependence of the sensor performance on the number of membranes, the hypothesis presented are not confirmed by some evidence or calculation.

Author Response

Thank you for the kind review of our manuscript. Authors also appreciate positive comment from the reviewer. The sensor performance on the membranes’ number was for the optimization of the fabricated sensor to monitor urea concentration in flow condition. The sensor performance was increase as the number of membranes increased and showed the maximum at 8 membranes. From these data, we confirmed that the optimal number of membranes was eight.

Reviewer 2 Report

The title of the manuscript need to be modified mentioning the specific word characterization, to make it more suitable t work and specific to the manuscript. To be specific, please mention the specific applications and don’t be too general. Highlights need to be rewritten for the effective communication. Comparison of similar kind of works to be done in forms of figures and tables with your findings in the manuscript, if applicable or else justifies. From the Fig. 3 – Scanning Electron Microscopy (SEM), it can’t be recognized the different functionalization of the sensor and even from the SEM image there is no changes have seen so claiming the functionalization by SE is just a imagination which incorrect. In this case author can do the elemental mapping to shoe the functionalization/sensor fabrication. In order to show the specificity of the fabricated sensor, author need to perform the interference study. The article reference are very old, this study is not supported by the current reference, author need to respond that what is reason behind this. Is this approach is very less studied and it’s a new report or author missed to cite the recent references. The author needs to add the more recent article in last five year, accordingly. Abstract of the manuscript is not well presented and its not covering all the point. Author need to rewrite the abstract in the way that abstract can present all the point, as author cover in the entire manuscript.

Author Response

The title of the manuscript need to be modified mentioning the specific word characterization, to make it more suitable to work and specific to the manuscript. To be specific, please mention the specific applications and don’t be too general.

=>We would like to thank the reviewer for thoughtful review of the manuscript. According to the reviewer’s comment, the title has been changed to “Parylene-Coated PTFE-Membrane-Based Portable Urea Sensor for Real-Time Monitoring of Urea in Peritoneal Dialysate”.

Highlights need to be rewritten for the effective communication.

=> According to the reviewer’s comment, highlights has been rewritten with better effective communication.

[Highlights]

Portable electrochemical urea sensor for real-time monitoring of urea in flowing peritoneal dialysate. Urease was immobilized on porous PTFE membranes via amine functionalized parylene-A coating and glutaraldehyde cross-linking. Simple urea monitoring without any secondary enzyme including its cofactor or complex nano-structure. The urea biosensor was confirmed to be suitable for the real-time monitoring of urea at flow rates ranging from 0.5 to 10 mL/min. The urea biosensor was suitable for the urea monitoring in the waste peritoneal dialysate of a patient with chronic renal failure.

Comparison of similar kind of works to be done in forms of figures and tables with your findings in the manuscript, if applicable or else justifies.

=> As reviewer’s comment, comparison of similar studies is very meaningful to prove the novelty of research. In this work, we focused on the fabrication of the urea biosensor for use in fast flow condition. There were several studies adopted the flow system in urea biosensor but no studies measured the urea concentration in flowing sample. Very recently, we have been reported the continuous detection of urea in the flow condition and published in Sensors. In that study, it was confirmed to be feasible at flow rate slower than 0.5 mL/min. So, there is no study to compare with our work and this state has been described in introduction part (page 2, line 61-80).

From the Fig. 3 – Scanning Electron Microscopy (SEM), it can’t be recognized the different functionalization of the sensor and even from the SEM image there is no changes have seen so claiming the functionalization by SE is just a imagination which incorrect. In this case author can do the elemental mapping to shoe the functionalization/sensor fabrication.

=> SEM instrument in authors’ laboratory has atomic analysis system, EDS and the atomic distribution of SEM samples were analyzed. The results of EDS has been described in table 1 and explanation also has been described in the modified manuscript.

[3.1. Fabrication of the urease immobilized PTFE membranes using a parylene coating, page 5, line 216-227]

The atomic distribution was analyzed using the energy dispersive spectrometry (EDS) in SEM. As shown in Table 1, Bare PTFE membrane showed only carbon (40.7 %) and fluorine (59.3 %). PTFE has four fluorine on saturated alkane chain so it is valid result. After parylene-A coating (AP membrane), nitrogen (9.8 %) appears and this nitrogen is from amine groups on parylene. When the AP membrane was treated with cross-linker, glutaraldehyde, additional oxygen (1.9 %) was observed. It means that AP membranes were reacted with oxygen containing glutaraldehyde. After urease immobilization, nitrogen and oxygen contents were increased to be 9.7 and 4.4 %, respectively. Since proteins like urease contains relatively higher amount of nitrogen and oxygen than organic polymers so this results means that urease has been immobilized on AP membrane via glutaraldehyde cross-linking. Based on these results, the fabrication of UAP membranes with an intact microporous structure and urease immobilization on AP membranes was confirmed.

[Added table, Table 1, page 6, line 234-236]

Table 1.Atomic distribution of bare PTFE membrane, parylene-A-coated PTFE membrane, glutaraldehyde-treated parylene-A-coated PTFE membrane, and urease-immobilized parylene-A-coated PTFE membrane. Numbers in table mean atomic %.

Sample

Carbon

Nitrogen

Oxygen

Fluorine

PTFE

40.7

-

-

59.3

AP

37.7

9.8

-

52.5

AP + glutaraldehyde

38.7

8.6

1.9

50.8

UAP

35.1

9.7

4.4

50.8

In order to show the specificity of the fabricated sensor, author need to perform the interference study. The article reference are very old, this study is not supported by the current reference, author need to respond that what is reason behind this. Is this approach is very less studied and it’s a new report or author missed to cite the recent references. The author needs to add the more recent article in last five year, accordingly.

=> According to the reviewer’s comment, recent references have been cited and the urease’s specificity has been described in the introduction part.

[Introduction, page 2, line 49-51]

Urease specifically reacts with urea so urease-based biosensor can minimize the interference from other molecules in physiological fluid such as uric acid, glucose and so on [15, 16].

[Added references]

Jakhar, S.; Pundir, C., Preparation, characterization and application of urease nanoparticles for construction of an improved potentiometric urea biosensor. Biosens. Bioelectron. 2018, 100, 242-250. Dervisevic, M.; Dervisevic, E.; Şenel, M., Design of amperometric urea biosensor based on self-assembled monolayer of cystamine/PAMAM-grafted MWCNT/Urease. Sensors Actuators B: Chem. 2018, 254, 93-101.

Abstract of the manuscript is not well presented and its not covering all the point. Author need to rewrite the abstract in the way that abstract can present all the point, as author cover in the entire manuscript.

=> According to the reviewer’s comment, the abstract has been rewritten.

[Abstract, page 1, line 16-30]

A portable urea sensor for use in the fast flow condition was fabricated using porous polytetrafluoroethylene (PTFE) membranes coated with amine-functionalized parylene, parylene-A, by vapor deposition. The urea-hydrolyzing enzyme urease was immobilized on the parylene-A-coated PTFE membranes using glutaraldehyde. The urease-immobilized membranes were assembled in a polydimethylsiloxane (PDMS) fluidic chamber, and a screen-printed carbon three-electrode system was used for electrochemical measurements. The success of urease immobilization was confirmed using scanning electron microscopy, and Fourier-transform infrared spectroscopy. The optimum concentration of urease for immobilization on the parylene-A-coated PTFE membranes was determined to be 48 mg/mL, and the optimum number of membranes in the PDMS chamber was found to be 8. Using these optimized conditions, we fabricated the urea biosensor and monitored urea samples under various flow rates ranging from 0.5 to 10 mL/min in the flow condition using chronoamperometry. To test the applicability of the sensor for physiological samples, we used it for monitoring urea concentration in the waste peritoneal dialysate of a patient with chronic renal failure, at a flow rate of 0.5 mL/min. This developed urea biosensor is considered applicable for (portable) applications such as artificial kidney systems and portable dialysis systems.